**METHOD**

# Modeling group heteroscedasticity in single-cell RNA-seq pseudo-bulk data

Yue You[1,2*] ⓘ, Xueyi Dong[1,2], Yong Kiat Wee[3], Mhairi J. Maxwell[3], Monther Alhamdoosh[3], Gordon K. Smyth[4,5], Peter F. Hickey[1,2,6], Matthew E. Ritchie[1,2*] and Charity W. Law[1,2*]

*Correspondence:
you.y@wehi.edu.au;
mritchie@wehi.edu.au;
law@wehi.edu.au

[1] Epigenetics and Development
Division, The Walter and Eliza Hall
Institute of Medical Research, 1G
Royal Parade, Parkville, Australia
[2] Department of Medical Biology,
The University of Melbourne,
Parkville, Australia
[3] CSL Limited, Parkville, Australia
[4] Bioinformatics Division, The
Walter and Eliza Hall Institute
of Medical Research, 1G Royal
Parade, Parkville, Australia
[5] School of Mathematics
and Statistics, The University
of Melbourne, Parkville, Australia
[6] Advanced Technology
and Biology Division, The Walter
and Eliza Hall Institute of Medical
Research, 1G Royal Parade,
Parkville, Australia

## Abstract

Group heteroscedasticity is commonly observed in pseudo-bulk single-cell RNA-seq datasets and its presence can hamper the detection of differentially expressed genes. Since most bulk RNA-seq methods assume equal group variances, we introduce two new approaches that account for heteroscedastic groups, namely *voomByGroup* and *voomWithQualityWeights* using a blocked design (*voomQWB*). Compared to current gold-standard methods that do not account for group heteroscedasticity, we show results from simulations and various experiments that demonstrate the superior performance of *voomByGroup* and *voomQWB* in terms of error control and power when group variances in pseudo-bulk single-cell RNA-seq data are unequal.

**Keywords:**  Pseudo-bulk scRNA-seq, Differential expression analysis, Group heteroscedasticity

## Background

Single-cell RNA sequencing (scRNA-seq) allows the quantification of transcript profiles across individual cells and has become widely adopted over the past few years. A major advantage of scRNA-seq is the high resolution it offers, enabling researchers to study molecular responses to different biological perturbations at the cellular-level [1] rather than the population-level as surveyed by bulk RNA-seq approaches. Many statistical tools and methods have been developed to make use of these high-resolution data, such as methods for trajectory analysis [2], cell-to-cell interactions [3], and differential expression (DE) analysis [4, 5].

Early DE analysis of this data type aimed to fully leverage information from individual cells, whereby each cell in comparison is treated as an independent biological unit (or "replicate"). To achieve this, a number of studies used established methods developed for bulk RNA-seq data [6]. However, due to the sparsity of the gene count matrix, which is a major point of difference between single-cell and bulk data [4, 5], other researchers modeled scRNA-seq data as either zero-inflated or multi-modal in distribution and

developed tailored DE analysis methods for scRNA-seq data (e.g., *MAST* [4], *BPSC* [7], and *DEsingle* [8]). To guide the analysts' choice, various evaluation studies have assessed the performance of bulk and tailored scRNA-seq analysis methods, although their findings have varied. Some showed that bulk methods are unsuitable when directly applied to scRNA-seq data [9, 10], while others found bulk methods were comparable to tailored scRNA-seq methods [11]. Another analysis strategy performs DE analysis on pseudo-bulk samples that are created by cell aggregation [12]. This strategy was pointed out to perform better than single-cell methods that treat each cell as an independent replicate in the analysis in two independent studies [13, 14]. Through the use of an aggregation approach, dependencies between cells from the same sample are avoided [15] so that the intrinsic variability of biological replicates is well-estimated leading to fewer false discoveries compared to methods that fail to account for this [14]. Although a generalized linear mixed model with a random effect to take care of zeros and correlation structure within a sample provides slightly more power compared to pseudo-bulk aggregation methods [13, 16], it brings a much heavier computational burden [13, 14].

Most of the DE analysis methods applied on pseudo-bulk data in the literature are "gold-standard" bulk DE analysis methods, including *limma-voom*, *limma-trend* [17, 18], *edgeR* [19], and *DESeq2* [20]. *Limma* was developed for microarray data, assumes the log-transformed expression values are normally distributed, and employs linear modeling and empirical Bayes shrinkage to improve the stability and power of statistical tests. For RNA-seq data, *voom* and *limma-trend* were subsequently developed based on the assumption of normality of the log-transformed counts, using different strategies to address the dependence that is observed between the mean and the variance (referred to as heteroscedasticity) in this kind of data. *Voom* models the relationship between the mean and variance across all observations using a fitted LOWESS trend and calculates precision weights based on the estimated trend for use in linear modeling. On the other hand, *edgeR* employs empirical Bayes shrinkage and was developed assuming gene-level counts follow a negative binomial distribution.

Due to limited sample numbers, most bulk DE analysis methods including the aforementioned gold-standard methods borrow information between genes to estimate the variance [19, 21–23] and assume equal variances between experimental groups (also referred to as "homoscedasticity"). However, there are cases where the variability observed is distinct for different groups ("heteroscedasticity"). Here we use "group" as a general term that covers common experimental variables or conditions such as treatment (drug A, drug B, vehicle control), genotype (wild-type, knock-out), and sex (male, female). In scRNA-seq analysis, DE methods can be used to find marker genes as well [24], in which case, the concept of "group" can extend to different cell types or clusters.

Heteroscedasticity has been frequently observed in microarray gene expression data [25, 26], for instance, Demissie et al. showed that a moderated Welch test performs better than the moderated *t*-test when group variances are unequal [26]. In large-scale bulk RNA-seq data, under the scenarios of heteroscedasticity, Ran et al. pointed out that *voom* was unable to model the variability appropriately and they noted that the weighting strategy used in *voomWithQualityWeights* (*voomQW*) may be more helpful [27] on account of its joint modeling of variability at the observational and sample-level. Chen et al. noted an unequal group variance in single-cell data as well, stating that unequal

variance tests are underused [28]. They made use of the large sample sizes available when each cell is considered as a replicate and estimated group-specific dispersions for each gene separately.

In this article, we examine whether group-specific variances are homoscedastic (equal) or heteroscedastic (unequal) in pseudo-bulk scRNA-seq data. We show that heteroscedastic groups are frequently observed in the data and that the application of current DE analysis methods has variable performance. Importantly, gold-standard methods that do not model group-level variability can both under- and over-estimate variances leading to poor error control or reduced power to detect DE genes. We demonstrate that methods that account for heteroscedastic groups, namely *voomByGroup* and *voomQW* using a blocked design, have superior performance in this regard when group variances are unequal.

## Results

### Observing heteroscedasticity in scRNA-seq pseudo-bulk data

To study whether group variances are equal or unequal in scRNA-seq pseudo-bulk data, we explored pseudo-bulk scRNA-seq datasets generated with cells from specific cell types obtained from various sample types ranging from experimental replicates of mice to human samples (see the "Methods" section). We examined three things: (1) multi-dimensional scaling (MDS) plots, (2) common biological coefficient of variation (BCV) of groups, and (3) mean-variance trends derived from individual groups (we refer to these as "group-specific *voom* trends"). Larger distances between samples in a group on the MDS plots indicate more within-group variation. BCV is a measure of the biological variability in gene expression between biological replicates and is frequently used to estimate the variance of gene expression in RNA-seq data [29]. Higher common BCV values correspond to increased biological variation between samples across genes based on the assumption of the negative binomial (NB) distribution in *edgeR*. For the group-specific *voom* trends, we are interested in observing where the curves sit relative to other groups in the same study, as well as the shape of the curve. The shape and "height" of the curves reflect the total variation within groups—both technical and biological.

In studies of mouse lung tissue [30] and *Xenopus* tail [31], we observed some minor differences in group-specific *voom* trends, with the curves sitting close together and mostly overlapping one another (Fig. 1a-b). Common BCV values for these studies ranged from 0.197 to 0.240 across 2 groups in mouse lungs, and 0.226 to 0.295 across 5 groups in *Xenopus* tails.

In human datasets, the differences in group variances were greater. For human peripheral blood mononuclear cells (PBMCs) [32], healthy controls unsurprisingly exhibited lower variability than the 3 other patient groups to which they were compared (Fig. 1c). This was evident from group-specific *voom* trends—although the curves had similar shapes, the curve of the healthy controls sat distinctly below the curves of other groups. Common BCV values for the PBMCs ranged from 0.154 to 0.241, with the lowest for healthy controls and the highest for asymptomatic patients.

A separate study on human macrophages collected from lung tissues [33] showed even higher levels of heteroscedasticity, where common BCV values ranged from 0.338 to 0.495 (Fig. 1d). Group-specific *voom* trends had distinct shapes and were well separated

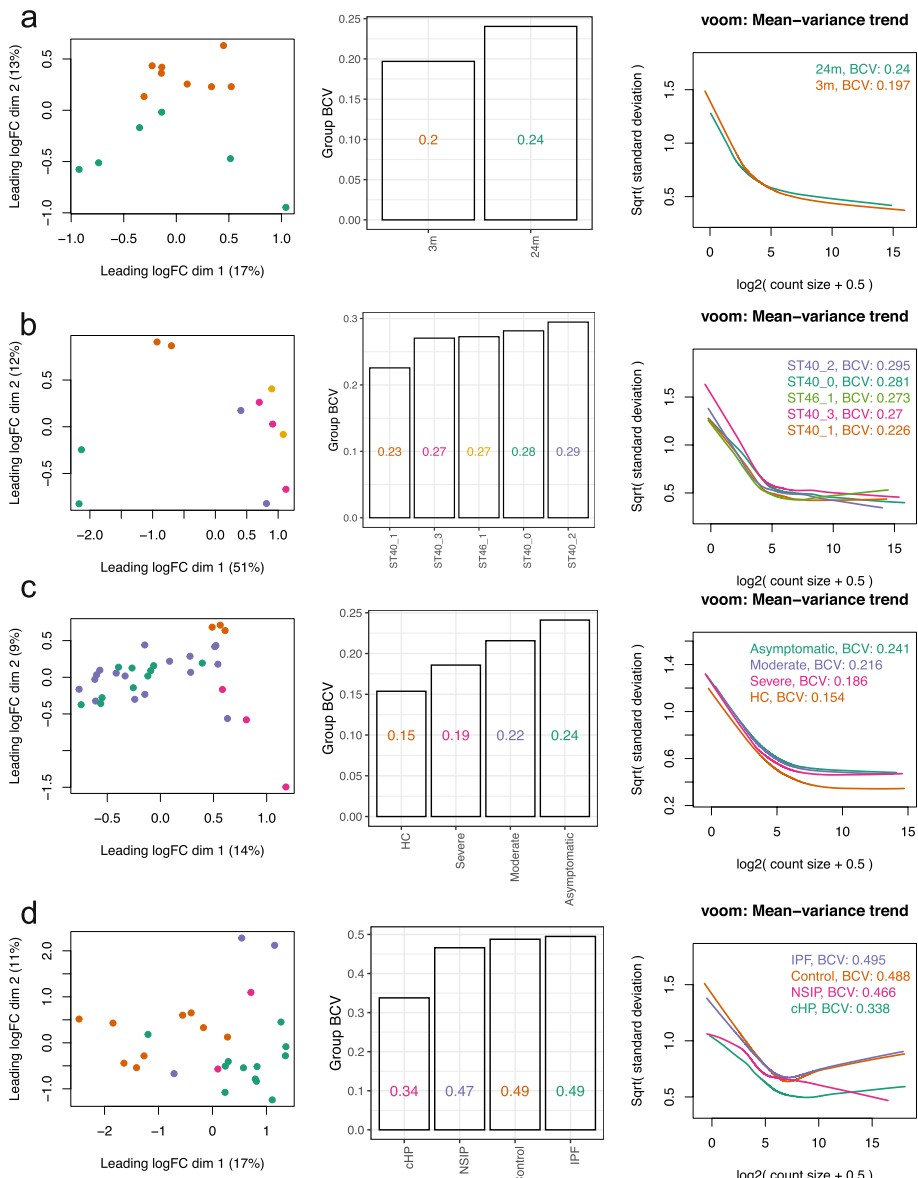

**Fig. 1** Group variation in scRNA-seq pseudo-bulk datasets. Group variation in 4 publicly available scRNA-seq datasets with various experimental designs, with replicates samples from **a** mouse lungs, **b** *Xenopus* tails, **c** human PBMCs, and **d** human lungs, is summarized. For each scRNA-seq dataset, cells of one particular type were selected (see the "Methods" section), and the cells from each sample were aggregated to create pseudo-bulk counts. Multidimensional scaling plots of pseudo-bulk data were plotted in the left panel, with distances computed from the log-CPM values and samples colored by groups. Group-wise common BCVs are plotted in the middle panel. Group-wise mean-variance trends are plotted in the right panel. Colors denote groups

from one another along the vertical axis (with the exception of IPF and control groups which were quite similar). Moreover, the plateauing of *voom* trends at higher expression values that are commonly observed in many datasets was not observed here. This might be on account of the complexity of regions in the lung where samples are collected, the diverse causes of lung fibrosis, and limited patient numbers for some groups. High levels of biological variation are reflected in the large BCV values in this dataset.

While we have not commented specifically on MDS plots, these plots (or other similar plots e.g., principle components analysis) provide a useful first glance of the data and are already part of many analysis pipelines (Fig. 1). For example, in the study of mouse lung tissue, the 3-month (3m) samples are less spread out across dimensions 1 and 2 than the 24-month (24m) samples, indicating that the 3m group has lower variability than the 24m group. This is confirmed by the groups' BCV values and *voom* trends.

In conclusion, we observe unequal group variability across multiple scRNA-seq pseudo-bulk datasets. At this stage, it is unclear whether gold-standard bulk DE analysis methods are robust against heteroscedasticity, and how different group variances need to be before it affects their performance. We test this in later sections of this article, using three gold-standard methods that do not account for heteroscedasticity and two novel methods that do.

### Novel use of *voomWithQualityWeights* using a block design (*voomQWB*)

The first method that accounts for group-level variability makes novel use of the existing *voomQW* method. The standard use of *voomQW* assigns a different quality weight to each sample, which then adjusts the sample's variance estimate—a strategy used to tackle individual outliers in the dataset. Rather than adjusting the variance of individual samples, we adjust the variance of whole groups by specifying sample group information via the `var.group` argument in the `voomWithQualityWeights` function. This produces quality weights as "blocks" within groups (identical weights for samples in the same group) and adjusts each group's variance estimate—we refer to this method as "*voomQW* using a block design," or simply *voomQWB*.

Figure 2a shows the estimated group-specific weights from *voomQWB* for a study comparing healthy controls to COVID-19 patients that are moderately sick and those that are asymptomatic [32]. Samples of moderately sick and asymptomatic patients have similar weights, just under 1; while the weights for healthy controls are above 1 (1.27). The sample weights are combined with observation-level weights derived from the overall mean-variance trend from *voom* (Fig. 2b). What this achieves in practice is an upshift of the *voom* trend for groups with sample weights below 1 (Fig. 2c pink and green curves), resulting in a higher variance estimate and a smaller precision weight for statistical modeling (see the "Methods" section). On the other hand, groups with sample weights that are greater than 1 have a down-shifted *voom* trend (purple curve), resulting in lower variance estimates and larger precision weights. There are a couple of things to note here: (1) the group-specific *voom* trends from *voomQWB* (Fig. 1c) are roughly parallel to the single *voom* trend (Fig. 1b), and 2) the group-specific trends shown here are created manually, not as an output of the `voomWithQualityWeights` function.

### *voomByGroup*: modeling observation-level variance in individual groups

As mentioned above, *voomQWB* models group-wise mean-variance relationships via roughly parallel trend-lines, which has the disadvantage of not being able to capture more complicated shapes observed in different datasets (Fig. 1). The second method we describe here, called *voomByGroup*, can account for such group-level variability with greater flexibility. *voomByGroup* achieves this by subsetting the data and estimating separate *voom* trends for each group. In other words, while *voomQWB* can shift the same

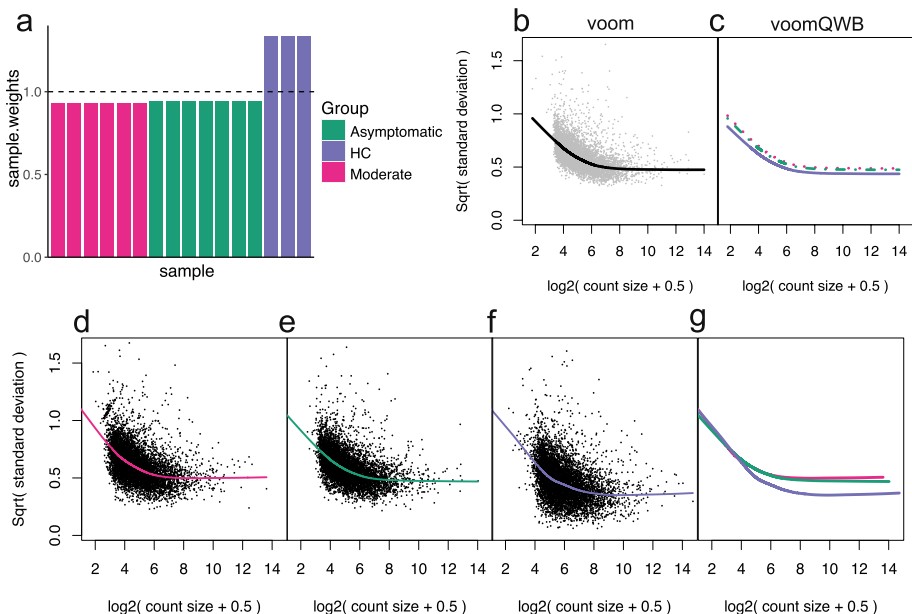

**Fig. 2** An overview of the *voom*-based mean-variance modeling methods applied. On the *PBMC1* pseudo-bulk data, group-specific weights estimated using voomQWB by defining each block (group) as different levels of the *symptom* variable are plotted in (**a**). The equal weight (= 1) level is plotted as a dashed line. Across all observations, gene-wise square-root residual standard deviations are plotted against average log-counts in gray in (**b**). *voom* applies a LOWESS trend (black curve) to capture the relationship between the gene-wise means and variances. Based on the final precision weights used in *voomQWB*, adjusted curves for each block are plotted in (**c**), where replicates in the same group share the same curve. Different colors and line types represent different groups (blocks). Dashed lines were used to avoid over-plotting. When *voomByGroup* is used, LOWESS trends are fitted separately to the data from individual groups to capture any distinct mean-variance trends that may be present (**d**–**f**). All group-specific trends from this dataset are plotted together in panel **g**, with different colors per group

*voom* trend up and down for each group, *voomByGroup* estimates distinct group-specific trends that can also allow up- and down-shifts for different groups.

For example, on the *PBMC1* dataset, the mean and variance are calculated for the $\log_2$counts-per-million (log-CPM) of each gene in "Group 1 (Moderate symptoms)" and a curve, or trend, is fitted to these values from which precision weights are then calculated (Fig. 2d). Similarly, a curve is fitted separately to each of the other groups in the dataset (Fig. 2e and f). This results in 3 non-parallel curves as shown in the summary plot (Fig. 2g) which includes all 3 trends. In theory, the *voomByGroup* method gives more robust estimates of variability since the trend for each group can take on a different "shape"—we test how this works in practice in the following sections.

### Group variance methods provide a balance between power and error control

Using simulated data, we test the performance of 3 gold-standard methods against the 2 new methods that account for heteroscedastic groups. The gold-standard methods are *voom*, *edgeR* using a likelihood-ratio test (*edgeR LRT*), and *edgeR* quasi-likelihood (*edgeR QL*). The methods that account for group heteroscedasticity ("group variance methods") are *voomQWB* and *voomByGroup*. Using simulations of pseudo-bulk data, we can examine the effects of unequal group variation while controlling other factors.

Specifically, group variation can be divided into biological variation between RNA samples and technical variation caused by sequencing technologies.

In the first scenario (*scenario 1*), we looked into unequal group variation as a result of biological variation. To obtain pseudo-bulk data, we simulated single-cell gene-wise read counts that followed a correlated negative binomial distribution and aggregated the reads from each sample (see the "Methods" section and Additional file 1: Fig. S1). Each simulation consists of 4 groups with 3 samples in each group—a total of 50 such simulations were generated. We generated varying group-specific common BCVs for the 4 groups that are well within the range of BCV values observed in experimental datasets (Figs. 1 and 3a)—the BCV values averaged over 50 simulations were 0.2, 0.22, 0.26, and 0.28 (the values in Fig. 3a are for one such simulation).

The mean-variance trends generated for the different groups appear as expected, with a typical decreasing "*voom*-trend" with increasing gene expression and the curves ordered correctly from those with the most biological variation at the top of the plot (group 4 in Fig. 3a) to the group with the least biological variation at the bottom of the plot (group 1). The left-hand side of these mean-variance trends is primarily driven by

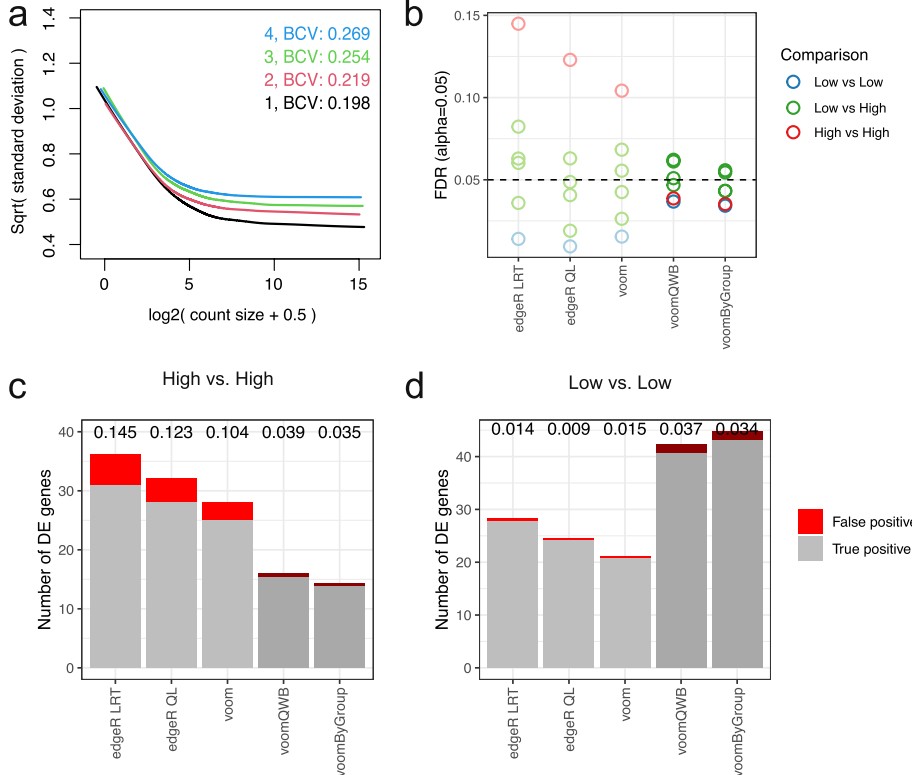

**Fig. 3** Group variance modeling methods provide good power while controlling the false discovery rate. **a** Mean-variance trends plotted for each group by the `voomByGroup` function on simulated scRNA-seq data with varying group-specific common BCV values, where colors represent different groups. In terms of the simulated variability-level, groups 1 and 2 represent *Low* variation, while 3 and 4 have *High* variation. Based on DE analysis results, FDR across methods in different comparisons (colors denote the 3 comparison types) are summarized in panel (**b**) at a cut-off of 0.05. The number of DE genes recovered by different methods for comparison between the two groups with higher variability (panel **c**) and the two groups with lower variability (panel **d**) at the same FDR cut-off are shown. For each bar in these plots, gray represents true positive genes, red represents false positive genes, and the FDR is labeled at the top of the bar

technical variation—as expected, here they mostly overlap each other since groups were generated to have the same technical variation in these simulations. These group-specific mean-variance plots generated by the `voomByGroup` function provide a useful "first glance" of the data before any formal testing was carried out.

We then performed differential gene expression analysis for all pairwise group comparisons, which gave a total of 6 comparisons. In these simulations, 50 genes were generated to be upregulated in each group, such that 100 genes are differentially expressed in each pairwise comparison (see the "Methods" section). We noticed that the number of differentially expressed genes varied from method to method and calculated the false discovery rate (FDR) of each method which was averaged over the 50 simulations. The FDR, or type I error rate, is calculated as the number of genes that were incorrectly identified as differentially expressed out of the total number of genes that were identified as differentially expressed at a particular adjusted *p*-value cut-off. We observed that none of the methods controlled type I error for all comparisons across the 3 cut-offs we examined: adjusted *p*-value cut-off of 0.01, 0.05, and 0.10 (Fig. 3b, Additional file 1: Fig. S2); such that the methods detected false discoveries at a higher rate than expected. *voomBy-Group* out-performed other methods by controlling type I error at the 0.01 and 0.10 cut-offs, and only exceeding the threshold marginally for 2 out of 6 comparisons at the 0.05 cut-off (FDR of 0.054 and 0.056).

A closer look at these plots revealed that the gold-standard methods had FDR values that spanned a broad range, with some comparisons having FDR values that were well under the threshold, and others that exceeded the threshold by 2- or 3-fold. This means that it could be difficult to gauge whether the DE results are too conservative, too liberal, or perhaps "just right" for a given comparison in real datasets when applying these methods to heteroscedastic groups. The range of FDR values is broadest for *edgeR LRT*, followed by *edgeR QL*, then *voom*. In comparison, the group variance methods, though not perfect in terms of type I error control, had a substantially tighter range of FDR values, and the comparisons that exceeded the FDR threshold only exceed it by a small margin.

To understand how heteroscedasticity influences DE analysis in more detail, we focused on results obtained using a 0.05 adjusted *p*-value cut-off. Across the 6 comparisons, group variance methods tend to detect similar numbers of differentially expressed genes, the same goes for gold-standard methods (Fig. 3c-d, Additional file 1: Fig. S3). There is some variation between the gold-standard and group variance modeling methods, with some comparisons having quite similar results, while others produce results that are very different. A closer examination of the comparison between group 3 and group 4 (which we refer to as "*High vs High*" in terms of biological variation) and group 1 and group 2 ("*Low vs Low*") shows where the 2 classes of methods differ.

In the *High vs High* comparison, gold-standard methods detect more DE genes than group variance methods. However, the DE genes contain a much higher proportion of false discoveries than it was controlled for. It is not as though gold-standard methods were prioritizing false-positive genes in terms of significance—it had similar numbers of true- and false-positive genes when looking at top-ranked genes (Additional file 1: Fig. S4). Rather, gold-standard methods had smaller adjusted *p*-values than group variance methods, allowing more genes detected at a certain cut-off (Additional file 1: Fig. S5a). By pooling variance estimates across all 4 groups, gold-standard methods

under-estimate the variances for groups 3 and 4, when in fact those groups have relatively high biological variation, resulting in poor type I error control. *voomByGroup* and *voomQWB* are more robust in their estimation of individual group variances, allowing them to maintain good type I error control.

In the Low vs Low comparison, all methods have good type I error control, with group variance methods detecting substantially more DE genes than gold-standard methods. Although top-ranked genes were yet again very similar (Additional file 1: Fig. S4), this time, gold-standard methods had larger adjusted *p*-values than group variance methods (Additional file 1: Fig. S5b), meaning that fewer genes were selected at given threshold. Here, the pooled variance estimates used by gold-standard methods resulted in an overestimation of variances for the two groups with relatively low biological variation (groups 1 and 2). In consequence, gold-standard methods suffered from a loss of power.

In the Low vs High comparison, there were no significant benefits in applying the group variance methods even though these methods provide more accurate variance estimates than the standard methods. When using *voom*, the overall variance trend that is applied to all samples and groups would under-estimate the variability of high variance groups, and over-estimate this for low variance groups. When pairwise comparisons are made between High and Low groups, the over- and under-estimated values balance each other out, giving results that are similar to the more precise estimates from *voomByGroup* and *voomQWB*.

These simulations demonstrated that in the presence of group heteroscedasticity, group variance methods have a good balance between controlling type I error and the power to detect DE genes. To ensure that the superior performance of group variance methods was due to group heteroscedasticity in the data, we separately simulated 50 null simulations (*scenario 2*) where all groups had equal underlying biological variation (see the "Methods" section). We observed similar numbers of true positives and false discoveries between gold-standard and group variance methods (Additional file 1: Fig. S6).

### *voomByGroup* captures both biological and technical variation well in individual groups

Systematic differences are commonly observed in the sequenced libraries of scRNA-seq data [34]. For example, gene counts vary between cells on account of limited starting material per cell and variations in technical efficiency. In addition, the number of cells detected in each sample is also variable. After aggregating cells, pseudo-bulk samples have library sizes that are more variable than that of bulk RNA-seq data—contributing to a major source of technical variation in pseudo-bulk data.

We explore the influence of unequal library sizes by varying the number of cells in each sample for a new set of simulations. Keeping the underlying biological variation constant between groups (homoscedasticity), we first vary the library sizes of samples. The provided number of cells are 250, 250, and 250 in group 1, 250, 200, and 200 in group 2, 250, 500, and 500 in group 3, and 250, 750, and 750 in group 4 (Fig. 4a). The expected library size for each cell remains constant (see the "Methods" section).

Under this scenario (*scenario 3*), the mean-variance trends generated appear to mostly overlap each other on the right-hand side as expected on account of equal group dispersions, while on the left-hand side, slight differences appear (Fig. 4b). DE analysis was then performed over 50 simulations and averaged FDR rates and

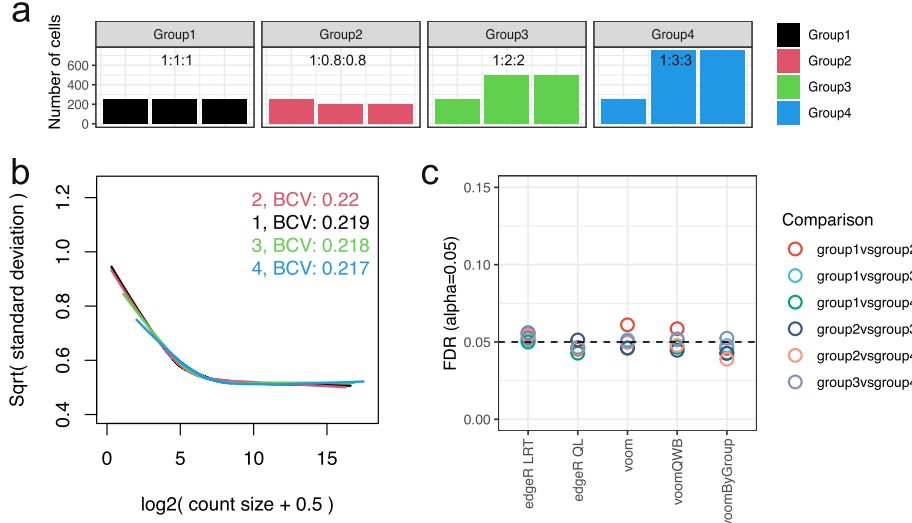

**Fig. 4** *voomByGroup* captures both biological and technical variation. **a** Summary of the simulation design with unequal numbers of cells per sample, with colors denoting the different groups in the dataset. In the scenario with technical variation only (unequal library sizes) across groups, mean-variance trends estimated by *voomByGroup* are plotted in panel (**b**), with group-wise common BCVs displayed in the top-right corner. Based on DE analysis results, FDRs across methods for different comparisons, denoted by distinct colors, are summarized in panel (**c**) at an FDR cut-off of 0.05

numbers of DE genes were compared. We observed much tighter ranges of FDR values compared to those from *scenario 1* (Fig. 4c, where the same *y*-axis range from Fig. 3b was used) and a similar number of true positive genes compared to the null simulation (Additional file 1: Fig. S7). These suggest that simulated technical variation does not have a significant influence on the DE results. To compare the influence of technical and biological variation in the simulations, we carried out another separate 50 simulations (*scenario 4*) with both aspects of variation incorporated (see the "Methods" section). For these results, we observed expected location trends that differed on both the left and right sides (Additional file 1: Fig. S8a). FDR results were rather similar to what was observed in the scenario where biological variation was unequal (Additional file 1: Fig. S8b, Fig. 3b), indicating that biological variation is the major source of variation that influences the DE results.

However, closer inspection of the FDR plot from *scenario 3* where only the number of cells differed between groups revealed that among those comparisons, *voom* and *voomQWB* performed similarly, as a result of the weighting strategy used in *voomQWB* only adjusting the group-wise weight in an overall manner. While *voomByGroup* is more flexible, we observed that for group-wise mean-variance trends, regardless of the overlapping trends on the right-hand side, on the left-hand side, groups with fewer cells (group1 and group2) exhibit slightly more variation and sit at the top, while the group with the largest number of cells (group4) is at the bottom (Fig. 4b). Because of the well-captured mean-variance relationship, *voomByGroup* delivered well-controlled FDR compared to those from *voom* and *voomQWB*, especially when comparing between group1 versus group2 in *scenario 3*, where slightly higher technical variation is present (Fig. 4c).

### Immune responses in asymptomatic COVID-19 patients

While the simulations allowed us to assess the performance of gold standard methods and group variance methods based on known truth, these results have no biological interest. Moreover, no matter how carefully thought-out and well-designed our simulations are, these data will inevitably miss some features from experimental data. Thus, we also examined the performance of methods on human scRNA-seq data.

Zhao et al. [32] investigated PBMCs from COVID-19 patients of varying severity alongside healthy controls (HCs), with a focus on the comparison between asymptomatic individuals and HCs. The study found that interferon-gamma played an important role in differentiating asymptomatic individuals and HCs, such that it was more highly expressed in natural killer (NK) cells of asymptomatic individuals [32]. In their data, the expression of *IFNG* was observed to be upregulated in asymptomatic individuals; however, the difference was not statistically significant when analyzed with *edgeR QL*. We reanalysed this dataset (*PBMC1*, see the "Methods" section) to see whether group variance methods could offer improved results.

We aggregated CD56$^{dim}$ CD16$^+$ NK cells from each sample to create pseudo-bulk samples and then filtered out samples with fewer than 50 cells. A first glance at the data via MDS and group-specific mean-variance plots shows that HCs have a distinct mean-variance trend and less biological variation (Fig. 1c). By accounting for the relatively low variance in the HC group, we found that group variance methods outperformed gold-standard methods in terms of statistical power, such that they detected more DE genes for the comparison between HCs and asymptomatic individuals (Fig. 5a)—this is consistent with our simulation results when comparing groups with low variance (Fig. 3b). *voomByGroup* detected the most DE genes, followed by *voomQWB*: 880 and 719 genes respectively. The gold-standard methods, *edgeR LRT*, *voom*, and *edgeR QL*, detected 664, 453, and 403 DE genes respectively.

To understand our results further, we looked at the consistency at which genes were detected as DE between methods (Additional file 1: Fig. S9a). We excluded *edgeR LRT* from our Venn diagram since the inclusion of all 5 methods greatly increased the complexity of the plot, and *edgeR LRT* was of less interest to us since we previously

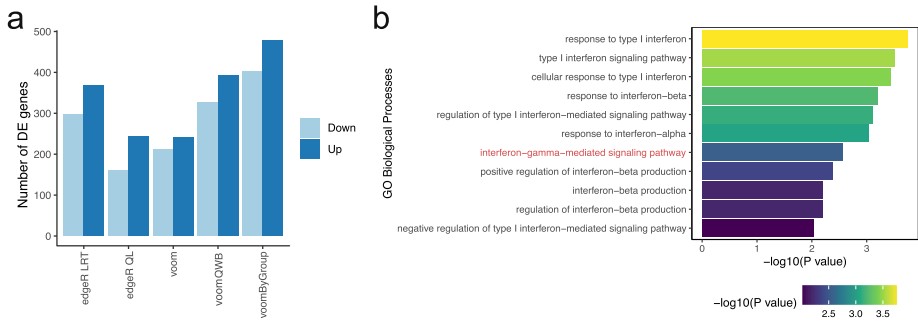

**Fig. 5** Genes differentially expressed between NK cells from asymptomatic COVID-19 patients and healthy controls. **a** The number of genes DE in the comparison between CD56$^{dim}$ CD16$^+$ NK cells from asymptomatic patients and healthy controls. Up means upregulated and down means downregulated in asymptomatic patients. Enriched GO terms related to interferon using DE genes detected with *voomByGroup* are plotted in panel (**b**). The *x*-axis displays the -log$_{10}$ transformed *p*-values for the different Gene Ontology terms, and the color-scale also varies by *p*-value

demonstrated that it performed poorly in the control of type I error. Although group variance methods detected almost double the number of DE genes as compared to *voom* and *edgeR QL*, most genes were detected by all methods (356 genes). Both of the *voom*-variants, *voomQWB* and *voomByGroup*, detected all of the genes that were also detected by *voom*. With the exception of 1 gene, *voomByGroup* also detected all of the genes that were detected by *voomQWB*. From *voom* to *voomQWB* then *voomByGroup*, the methods increase in their level of group-specific variance modeling. The overlap between these methods and the extra DE genes reflects the hierarchy in variance modeling for these methods and demonstrates the potential gain in statistical power when capturing group variance more accurately.

Next, we turned to Gene Ontology (GO) enrichment analysis to study the biological processes that play a role in COVID-19. We looked for any enrichment in GO terms for significantly upregulated genes in asymptomatic patients for each of the methods under examination. *voomByGroup* was the only method to detect the "interferon-gamma-mediated signaling pathway" as significant using a *p*-value cut-off of 0.01 (Fig. 5b). None of the other methods found any of these 5 genes as significant—they had much higher *p*-values as compared to that of *voomByGroup* (Additional file 1: Fig. S9b). To confirm the role of interferon-gamma in asymptomatic patients, we analyzed data from a separate study also involving $CD56^{dim}$ $CD16^+$ NK cells in COVID-19 patients of varying severity [35]. The original study did not look into the role of interferon-gamma. Reanalyzing these data (*PBMC2*, see the "Methods" section), we found that in this second dataset the "interferon-gamma-mediated signaling pathway" was enriched using any of the DE methods under examination, and those group-specific variances were similar between all groups (Additional file 1: Fig. S10).

Taking the two COVID-19 datasets into consideration, we noticed a few things: (1) group variances can change between one dataset and another, even for studies on similar cell types and similar subjects—this perhaps has to do with how samples are processed (technical variation) and/or the "grouping" criterion plus the individual subjects involved (biological variation); (2) when variance trends are not too distinct from one another, all methods perform similarly, as observed in the second dataset (*PBMC2*); (3) when variance trends are distinct, group variance methods may benefit from a gain in statistical power, as observed in the first dataset (*PBMC1*); and (4) by modeling group-specific variances closely, *voomByGroup* was the only method that obtained statistically significant results for the biological process of interest in both datasets. These two datasets are used here to highlight how results of biological interest may be "missed" if heteroscedasticity is not carefully considered.

## Discussion

We have shown that modeling the mean-variance relationship at the group-level and the use of group-wise precision weights enhances DE analysis results when there is group heteroscedasticity. Simulations demonstrated that *voomQWB* and *voomByGroup* have a good balance between controlling type I errors and the power to detect DE genes. Additionally, *voomByGroup* performs better at capturing technical variation in the mean-variance trends. The analysis of PBMC data agreed with our simulation results whereby methods that model group-specific variation provide more DE genes when low-BCV

groups are included in the comparison, with statistically significant results obtained for key biological processes of interest with *voomByGroup.* Null simulations confirmed that established gold-standard methods and approaches that model group-specific variation performed similarly when there were no distinct differences in variability between groups. Consistent results were presented by Chen et al. on scRNA-seq data [28] where methods that accounted for heteroscedasticity performed as well as methods that do not account for heteroscedasticity when there is equal group variation, which indicates there is potential for group-variation methods to be more broadly used.

In this article, we demonstrate that group variance modeling methods outperform gold-standard methods for DE analysis of pseudo-bulk scRNA-seq data. Specifically, *voomByGroup* has the best performance in terms of balancing type I error control and power. This is because *voomByGroup* models the mean-variance relationships for different groups more flexibly to better capture the distinct trends that may be present in the data. *voomQWB* also performs very well; with better results than gold-standard methods in the presence of group heteroscedasticity. Its performance is similar, and second only, to *voomByGroup*. Relative to *voomByGroup*, *voomQWB* lacks the flexibility to capture the distinct shapes of group-specific mean-variance trends, which could explain some of the differences in performance.

We recommend the use of either *voomByGroup* or *voomQWB* over gold-standard methods in scRNA-seq pseudo-bulk analysis in datasets that exhibit heteroscedastic variation across different experimental groups. The *voomByGroup* software provides useful diagnostic plots that can help guide the choice of method, with code that is easy to run, taking similar inputs to the widely used and well-established *voom* approach (see the "Methods" section). Running *voomByGroup* first can allow the analyst to determine the level of heteroscedasticity in a given dataset. For example, if the mean-variance trends per group are mostly overlapping each other, then group variance methods are likely to offer very similar results to current gold-standard methods (Fig. 4b-c). In this case, method choice will not affect the results much, and one may prefer to choose a method that is simpler, based on fewer assumptions, such as *voom*. If *voomByGroup* mean-variance plots show distinct trends in one or more groups, then the variance for that group can be more closely modeled using *voomByGroup* or *voomQWB* (e.g., "ST40_3" in Fig. 1b, "HC" in Fig. 1c, and "NSIP" and "cHP" in Fig. 1d). In such cases, methods that explicitly model group-specific variability are highly recommended over standard methods that do not. Moreover, the common BCV values that are automatically generated and displayed on these plots provide summary information about differences in mean dispersion for different groups calculated across all genes.

Between the two group variance methods, *voomByGroup* out-performs *voomQWB* slightly. It also provides group-specific mean-variance plots that are a useful diagnostic in exploratory data analysis. *voomByGroup*, however, has some limitations related to its use of a subset of the design matrix and data—a necessary step to obtain distinct group-specific shapes for the mean-variance trends. This means that in practice, the use of *voomByGroup* is most suitable for simple block designs with a single group factor only [36]. When there are additional explanatory variables, *voomByGroup* may not estimate covariates accurately or may run out of degrees of freedom when estimating coefficients for additional factors. For these complex experimental designs,

*voomQWB* is ideal since it can handle the same complexity as gold-standard methods such as *voom*, but with the additional safeguard against heteroscedastic groups. One such example of this includes datasets that are collected over several batches. *voomQWB* can properly accommodate biological groups of interest and batch information into the linear modeling, while handling differences in group variability.

For experiments with very small group sizes, *voomByGroup* offers the option of applying the overall *voom* trend to specific groups rather than using the default group-specific trend—this is specified in the `dynamic` argument of the function. Since *voomByGroup* estimates group variances using only the relevant samples, group-specific mean-variance trends could be unstable when modeled using a limited number of samples. We recommend the application of an overall *voom* trend to groups of size 1 or 2.

In situations where there are individual samples with higher variability (outliers), the *voomQWB* and *voomByGroup* methods may work less well, with the inclusion of highly variable samples increases the estimated group variation, which may decrease power. In these situations, regular sample-specific modeling of variability (i.e., *voomWithQualityWeights* without specifying the `var.group` option) would be more appropriate. In our study, we did not explore datasets with outlier samples and leave such investigations as future work.

In this study, we also observed that *edgeR*-based methods returned relatively different results compared to *voom*-based methods (Fig. 5a). One major source of this is the different distributional assumptions between methods. Due to the mathematical intractability of the NB distribution (basic distribution in *edgeR*) compared to the normal distribution, methods were first developed for modeling group heterogeneity in *limma* (e.g., *voomWithQualityWeights*), which assumes normally distributed data. When modeling data using a NB-GLM, modeling group-wise variation is more challenging. An example of weighted regression in this context comes from Zhao et al. [37], who used observational weights to account for outlier observations.

In our article, we focus on DE analysis of scRNA-seq pseudo-bulk data because recent benchmark studies have shown that it gives better results relative to analyzing scRNA-seq data in its original non-aggregated form [13, 14, 38]. However, it is worth noting that by aggregating single-cell data to obtain pseudo-bulk samples, the variance between cells of the same sample is masked. Thus, it may be useful to check cell-level gene expression and its variability, especially for any genes that are detected as significant. To account for this, Zimmerman et al. modeled the correlation structure between cells using a generalized mixed model where individuals were assigned as a random effect [16]. A similar approach was taken in Crowell et al. [13]. In a similar way, linear mixed modeling may also be accessible by using the *voomQWB* method together with the `duplicateCorrelation` function in the `limma` package.

Whilst we apply group variance methods on pseudo-bulk samples in this article, the idea of modeling group variances more closely can in theory be extended to DE analysis of other data types such as bulk RNA-seq data, pseudo-bulk of spatial scRNA-seq data, and surface protein data from CITE-seq. Moreover, the "groups" that are used by *voomQWB* or *voomByGroup* can be extended to cell types or clusters to find marker genes.

## Conclusion

In the presence of group heteroscedasticity, the *voomQWB* and *voomByGroup* methods have superior performance to approaches that do not account for distinct group-specific variation. These methods offer a better balance between false discovery control and the power to detect DE genes on pseudo-bulk scRNA-seq datasets. We recommend both group variance modeling methods, with *voomByGroup* having accurate variance estimation for simple designs, and *voomQWB* capable of modeling data with more complex study designs. To guide an analyst's choice of the most appropriate variance modeling method to apply to a given dataset, we recommend checking the relevant diagnostic plot to assess whether the model assumptions are met.

## Methods

### Revisiting variance modeling with *voom*

The group variance methods presented in this article, *voomQWB* and *voomByGroup*, are adaptations of *voom* method by Law et al. [18]. Briefly, the *voom* method fits a linear model to each gene using a design matrix with full column rank, *X*, such that

$$E(y_g) = X\beta_g,$$

where $y_g$ is a vector of log-CPM values for gene *g*, and $\beta$ is a vector of regression coefficients for gene *g*. The fitted model allows us to calculate residual standard deviations $s_g$. Square-root standard deviations $\sqrt{s_g}$ are plotted against the average log count of each gene, and a LOWESS curve [39] is fitted to the points—this creates the *voom*-style mean-variance plots seen throughout this article (Fig. 2b). Precision weights $w_{gi}$ for gene *g* and sample *i* are then calculated as a function of the fitted counts $\hat{\lambda}_{gi}$ using the LOWESS curve, such that $w_{gi} = \text{lo}(\hat{\lambda}_{gi})^{-4}$. The weights $w_{gi}$ are then associated with log-CPM values $y_{gi}$ in the standard *limma* pipeline, which uses these in weighted least squares regression.

### Group variance modeling with *voomQWB*

Written with outlier sample detection in mind, Liu et al. [40] combined sample-specific weights with the *voom* precision weights in their *voomWithQualityWeights* method. The combined weights, denoted as $w_{gi}^*$, can be described as $w_{gi}^* = w_{gi}/\exp\hat{\gamma}_i$, where $1/\exp\hat{\gamma}_i$ represents the sample-specific weights. The standard use of *voomWithQualityWeights* calculates sample-specific weights based on the similarity of gene expression profiles within groups, such that any sample that is dissimilar to the rest of the samples in the same group gets down-weighted. In other words, samples within the same group can be assigned varying weights.

Instead, we force samples within the same group to have the same weight by exploiting the `var.group` argument in the `voomWithQualityWeights` function. A factor representing the groups `group` is assigned to `var.group` to obtain "blocked" weights for the samples. Visually, what this achieves is an adjustment of the standard *voom* curve to separate curves for each of the groups, where the adjustment is based on the blocked sample-specific weights (Fig. 2c). In practice, the blocked

sample-specific weights are used to adjust the precision weights fed into the standard *limma* pipeline, such that $w_{gi}^*$ rather than $w_{gi}$ is used.

### Group variance modeling with *voomByGroup*

Our second group variance method, *voomByGroup*, tackles heteroscedastic groups from a different angle. *voomByGroup* subsets the gene expression data and design matrix $X$ for each group, such that a LOWESS curve is created using only the data from specific samples. The LOWESS curve is then used to obtain precision weights $w_{gic}$ for gene $g$ and sample $i$ in a group (or condition) $c$. As a result, each group has its mean-variance curve and set of weights (Fig. 2d-g). The group-specific weights are combined across all groups, $c = 1, 2, ...C$, to get $w_{gi}^\#$ which replaces $w_{gi}$ in the standard *limma* pipeline. Since *voomByGroup* subsets the data and the design matrix $X$ to obtain precision weights, any additional covariates are estimated using only a subset of the data.

Additionally, *voomByGroup* offers an option to make the usage of overall *voom* trend instead of group-specific trends, which is specified in the argument `dynamic` with the input as a vector of BOOLEAN variables. The `dynamic` is recommended to be turned on for groups with small group sizes, e.g., 2 or fewer samples in a group. For groups with relatively more samples (3 or more than 3), the `dynamic` can remain FALSE, which means that to estimate the variance, the group-specific trends are still used.

### Running variations of *voom*

The *voom*, *voomByGroup*, and *voomQWB* methods are run in `R` using the following functions:

voom(y, design=design, ...)
voomByGroup(y, design=design, group=group, ...)
voomWithQualityWeights(y, design=design, var.group=group, ...)

All functions are run similarly, with 2 common arguments and an additional argument for *voomByGroup* and *voomWithQualityWeights*. `y` represents pseudo-bulk count data with $N$ samples and $G$ genes. `design` is the design matrix with $N$ rows matching the number of samples and $P$ model parameters. `group` is a factor vector that is of length $N$.

Group-specific mean-variance plots are produced in the `voomByGroup` function, by specifying plot="separate" to get individual mean-variance plots for each group (Fig. 2d-f) or plot="combine" to show all mean-variance curves in a single plot (Fig. 2g) which makes relative differences between the curves easier to spot. The common BCV values calculated using `estimateCommonDisp` function in *edgeR* are automatically added to the plots.

### DE analysis with *edgeR*

Besides the standard *voom* method, two further options for DE analysis using *edgeR*, namely *edgeR LRT* (likelihood-ratio test) [29] and *edgeR QL* (quasi-likelihood) [41], were also evaluated. To run *edgeR LRT*, `glmFit` was used with default settings, and only the count matrix and design matrix were provided, followed by `glmLRT`. To run *edgeR QL*, `glmQLFit` was used with default settings, and only the count matrix and design matrix were provided, followed by `glmQLFTest`. All genes with associated *p*-values from the DE test used were then extracted with the `topTag` function.

### Simulated scRNA-seq data

Single-cell gene-wise read counts were generated to follow correlated negative binomial distributions (Additional file 1: Fig. S2). Baseline expression frequencies were generated by the function `edgeR::goodTuringProportions` on reference data [42] (iTreg cells were used, and genes with UMI counts > 200 were kept). The expected library size for each cell is estimated using a log-normal distribution [43]. Parameters (location mu and scale sigma) are estimated based on the reference data as well, and they are then used to calculate expected library sizes. Then baseline proportions were multiplied by expected library sizes to generate expected read counts.

Read counts from the same subject were generated to be correlated using a copula-multivariate normal strategy. First, multivariate normal deviates were generated with the specified intra-subject correlation. Then, the normal deviates were transformed to gamma random variables by quantile-to-quantile transformations to represent the "true" expression levels of each gene in each cell. Then, Poisson counts were generated with expected values specified by the gamma variates. Here the gamma deviates represent biological variation between subjects and cells while the Poisson counts represent technical variation associated with sequencing [29]. This process ensured that the counts follow marginal negative binomial distributions and that counts for each subject are correlated. Importantly, the intra-subject correlation affects only the biological part of the variation whereas the technical variation remains independent.

The relationship of the dispersion of aggregated cells to the dispersion of single cells is approximate:

$$\phi_{agg} = correlation \times \phi_{sc} + (1 - correlation)/N$$

where $\phi_{agg}$ is the dispersion of aggregated pseudo-bulk data, *correlation* is the intra-subject correlation, and $\phi_{sc}$ is the dispersion of single-cell data. $N$ is calculated by $N = \frac{(\sum L_i)^2}{\sum L_i^2}$, where $L_i$ is the cell-wise library size. In the current study, intra-block correlation is set as 0.1 for all simulations.

### Simulation scenarios

In this study, we simulate data for 4 scenarios: (1) groups having different biological variation, (2) no biological or technical variation between samples or groups, (3) samples having different cell numbers (this induces differences in technical variation by having unequal sample sizes), and (4) samples having different cell numbers and groups having different levels of biological variation. We generate 50 simulations for each scenario, with each simulation involving 12 samples (4 groups, with triplicate samples in each) and 10,000 genes. To induce differentially expressed genes in the datasets, 50 genes are randomly selected to be upregulated in each group with a true $\log_2$-fold-change of 2. This means that for every pairwise comparison, there are 100 true DE genes. For each simulated dataset, genes with fewer than 30 reads across all pseudo-bulk samples were filtered out before DE analysis.

In the first scenario (*scenario 1*), we keep the expected library size of each sample consistent by generating 250 cells for each sample. By varying the dispersion in our

simulation, we obtain common BCV values that are variable between groups – 0.20, 0.22, 0.26, and 0.28.

The second scenario (*scenario 2*) generates homoscedastic groups, such that there should be no true differences (biological or technical) in group variability. We use the data here to confirm whether the methods behave in the way we would expect—that *voomQWB* and *voomByGroup* perform similarly to *voom* if there are no group differences. Here, each sample contains 250 cells and groups have BCV values of ≈0.22.

In the third scenario (*scenario 3*), the biological variation is consistent between groups (common BCV=∼0.22), but the number of cells varies for each sample. With the baseline number of cells set as 250, the samples are adjusted by these proportions: 1:1:1, 1:0.8,0.8, 1:2:2, and 1:3:3. The expected library for each cell remains constant, such that a sample with more cells is expected to have a library size that is proportional to its cell number.

The fourth scenario (*scenario 4*) combines elements from the first and third scenarios. Biological variation is adjusted such that group 1 and group 2 have less biological variation (common BCV=∼0.22), and group 3 and group 4 have relatively more biological variation (common BCV=∼0.24). Samples have variable cell numbers—generated using the same baseline cell number and proportions as for our third simulation.

### scRNA-seq datasets

Publicly available scRNA-seq datasets that were examined in this article in Fig. 1 include:

- Whole lung tissue from 3-month and 24-month-old mice [30]. Pseudo-bulk data from type 2 pneumocytes were created. These data are available from GEO under accession number GSE124872.
- *Xenopus* tail from regeneration-competent and incompetent tadpoles, 1–3 days post-amputation [31]. Pseudo-bulk data from goblet cells were created. The data is available in the scRNAseq package [44].
- PBMCs from healthy controls and COVID-19 patients of varying severity (asymptomatic, moderate, or severe) [32]. Pseudo-bulk data from CD56$^{dim}$ CD16$^+$ NK cells were created. These data are available from the CNGB Nucleotide Sequence Archive (CNSA) under accession number CNP0001250.
- Human lung tissue from non-fibrotic and pulmonary fibrosis lungs [33]. Pseudo-bulk data from macrophage cells were created. These data are available from GEO under accession number GSE135893.

### COVID-19 datasets: *PBMC1* and *PBMC2*

We examined two separate scRNA-seq datasets that sequenced PBMCs from COVID-19 patients with varying severity (asymptomatic, moderate, and severe) and healthy controls. We refer to the first dataset described above as "*PBMC1*". The second dataset, which we refer to as "*PBMC2*" [35], is available from https://covid19cellatlas.org/.

### Filtering, data normalization, and downstream analysis

Prior to creating pseudo-bulk samples, we performed filtering at the gene- and cell-level. We then selected one cell type per dataset to create pseudo-bulk samples. We filtered out pseudo-bulk samples with relatively fewer cells or smaller library sizes before performing DE analysis (see Additional file 2: Table S1 for further details).

Normalization was then performed for each dataset using the trimmed mean of M values (TMM) method [45] using the `calcNormFactors` function.

The `goana` function in *limma* was used to carry out Gene Ontology (GO) analyses on DE results from the COVID-19 NK cells, using the `org.Hs.eg.db` annotation package (version 3.14.0) [46].

### Software

Results were generated using R version 4.1.3 [47], and software packages *limma* version 3.50.0, *edgeR* version 3.36.0, and *ggplot* version 3.3.5 [48].

### Supplementary Information

---

**Additional file 1:** Supplementary Figures.

**Additional file 2:** Table S1. Summary of filtering steps applied to each dataset.

**Additional file 3:** Review history.

---

**Peer review information**

**Review history**
The review history is available as Additional file 3.

**Authors' contributions**
Y.Y. performed the data analysis, generated all figures, and wrote the manuscript. C.W.L., X.D., and Y.Y. created new analysis methods and code. Y.K.W., M.J.M., and M.A. provided relevant human datasets and feedback on analysis results. G.K.S. created the framework for data simulation and advised on methods development. P.F.H., M.E.R., and C.W.L. designed the study. G.K.S., P.F.H., M.E.R., and C.W.L. supervised the analysis and wrote the manuscript. All authors read and approved the final manuscript.

**Funding**
This work was supported by the Chan Zuckerberg Initiative Essential Open Source Software for Science Program (Grant no. 2019-207283 to G.K.S and C.W.L. and Grant no. 2021-237445 to G.K.S) and the Chan Zuckerberg Initiative DAF, an advised fund of Silicon Valley Community Foundation (Grant No. 2019-002443 to M.E.R.) and an Australian National Health and Medical Research Council (NHMRC) Investigator Grant (2017257 to M.E.R).

**Availability of data and materials**
Datasets of whole lung tissue from 3-month and 24-month-old mice [30] are available from GEO under accession number GSE124872. Datasets of Xenopus tail from regeneration-competent and incompetent tadpoles are available in the `scRNAseq` package [44]. Datasets of PBMCs from healthy controls and COVID-19 patients of varying severity are available from the CNGB Nucleotide Sequence Archive (CNSA) under accession number CNP0001250 [32], and https://covid19cellatlas.org/ [35]. Human lung tissue from non-fibrotic and pulmonary fibrosis lungs [33] are available from GEO under accession number GSE135893.
 Code for the ***voom***-based methods, simulations, and comparisons of DE analysis methods on the various datasets explored are available from GitHub at https://github.com/YOU-k/voomByGroup [49]. A Zenodo repository for the source codes is available under doi: 10.5281/zenodo.7847793 [50]. The source code is licensed under an MIT License.

### Declarations

**Ethics approval and consent to participate**
Not applicable.

**Competing interests**
The authors declare that they have no competing interests.

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

## 

