## [**Additional file 3:** Review history. · Genome Biology]

Review History

First round of review

Reviewer 1

Were you able to assess all statistics in the manuscript, including the appropriateness of statistical tests used? Yes, and I have assessed the statistics in my report.

Were you able to directly test the methods? No.

Comments to author:

The authors consider analysis of scRNA-seq data using pseudobulking and the limma-voom method. They highlight the case where different groups of samples may have different levels of biological variability (group-level heteroscedasticity). In limma, which uses weighted linear models, this may be accounted for by a choice of weights inversely proportional to the variance of noise in the data.

There are several important contributions:

* The necessity of considering group-level heteroscedasticity is demonstrated with real data. There really is variable biological variability between groups in scRNA-seq datasets.

* The effectiveness of two proposed methods at controlling the False Discovery Rate while providing good statistical power is demonstrated with simulated pseudobulk data. A new biological finding is demonstrated in one real dataset that is made possible by the new voomByGroup method.

* The ability of voom methods to cope with differing numbers of cells in different samples is demonstrated in simulated pseudobulk data. This is a question I have had, and I was pleased to see it addressed in this paper.

* A new voom function, voomByGroup, is provided. This is a useful addition to the voom functions already available in limma. There was already a similar function for microarrays, voomaByGroup, but not for count data. This function is currently available from a GitHub repository written for this paper. I presume the intention is to add it to the limma package, and that it will be useful for both bulk and single-cell data. The function also provides a diagnostic visualization to check if group-level heteroscedasticity needs to be accounted for.

Comments and questions:

Is there a theoretical or practical reason not to routinely apply voomWithQualityWeights per-sample rather than per-group? Per-sample weighting should automatically include any necessary per-group weighting.

Page 3 lines 29-31, sentence "For studies involving mice...", the phrasing is odd.

This is an incremental extension to previous work on limma, voom, and edgeR. Someone unfamiliar with these methods might find it hard going. For example, what is the quantity "Biological Coefficient of Variation"? There is a nice description of the voom method and variants in the "Methods" section of the paper that would make the results section clearer if read first. The introduction and results sections could refer confused readers to specific relevant sections in the methods.

Are the methods appropriate to the aims of the study, are they well described, and are necessary controls included? If not, please specify what is required.

Yes. Using real and simulated data, the authors convincingly demonstrate the necessity and effectiveness of the methods they describe.

Are the conclusions adequately supported by the data shown? If not, please explain

Yes. The paper is describing a new method applied to existing datasets published elsewhere. A variety of datasets are examined.

Are sufficient details provided to allow replication and comparison with related analyses that may have been performed? If not, please specify what is required.

Yes. The methods are well described and there is source code provided in a GitHub repository.

Does the work represent a significant advance over previously published studies?

Yes. See the contributions described above.

Is the paper of broad interest to others in the field, or of outstanding interest to a broad audience of biologists?

Pseudobulk differential expression testing is an important method for scRNA-seq data, because it can correctly allow for biological variability between samples while also being extremely computationally efficient. This paper provides guidance on doing this using the limma package and the new voomByGroup function, and highlights the need to think about group heteroscedasticity. This will be of broad interest to people working with scRNA-seq data.

Reviewer 2

Were you able to assess all statistics in the manuscript, including the appropriateness of statistical tests used? Yes, and I have assessed the statistics in my report.

Were you able to directly test the methods? No.

Comments to author:

In this article, two new approaches voomQWB and voomByGroup are introduced to account for group variance-heteroscedasticity in DE analysis using scRNA-seq data. The motivation is very clear: the existing methods ignoring such heteroscedastic can substantially inflate type-I error rate in DE analyses. The proposed approaches and comparison methods are well described and

are easy to follow. The proposed approaches `voomQWB` and `voomByGroup` have been shown to outperform the gold standard methods `edgeR` and `limma` in both simulation studies and real data applications in the presence of group heteroscedasticity, so they could be competitors in DE analyses using scRNA-seq data. Nevertheless, I have some concerns as stated below.

1. `voomQWB` applies the existing R function `voomWithQualityWeights` in the R package `limma`, while `voomByGroup` simply extends `voom` by allowing group-specific variances. Therefore, the novelty of the two approaches is quite limited. The major contribution of this article is its empirical findings: `voomQWB` and `voomByGroup` could outperform the gold standard methods `edgeR` and `limma` in both simulation studies and real data applications in the presence of group heteroscedasticity.
2. In the homoscedastic-group situation, `limma` is generally more powerful than `voomQWB` and `voomByGroup` with comparable type-I error rates. This is because `voomQWB` and `voomByGroup` need to estimate more variance parameters than `limma`. Therefore, `voomQWB` and `voomByGroup` are preferred only when group variances differ to some extent. This raises a question: how to choose a suitable approach in practice? For example, can the authors give a cut-off of the absolute value of the BCV difference between two groups? So that `voomQWB` and `voomByGroup` are preferred against `limma` and `edgeR` if and only if the difference exceeds the cut-off.
3. The simulation studies only cover low vs low and high vs high comparisons. What about low vs high comparison?
4. Only a small group size of 3 was considered in the simulation studies. It would be of interest to consider larger sample sizes to cover more real-world situations.
5. The numerical studies did not include the existing DE analysis methods specifically tailored for scRNA-seq data, such as MAST, BPSC, and DEsingle. Why?
6. A couple of typos: In Supplementary Figure S9(a) and the corresponding caption, `VQWB` and `voomQW` should read `voomQWB`.

Reviewer #1:

The authors consider analysis of scRNA-seq data using pseudobulking and the limma-voom method. They highlight the case where different groups of samples may have different levels of biological variability (group-level heteroscedasticity). In limma, which uses weighted linear models, this may be accounted for by a choice of weights inversely proportional to the variance of noise in the data.

There are several important contributions:

- * The necessity of considering group-level heteroscedasticity is demonstrated with real data. There really is variable biological variability between groups in scRNA-seq datasets.
- * The effectiveness of two proposed methods at controlling the False Discovery Rate while providing good statistical power is demonstrated with simulated pseudobulk data. A new biological finding is demonstrated in one real dataset that is made possible by the new voomByGroup method.
- * The ability of voom methods to cope with differing numbers of cells in different samples is demonstrated in simulated pseudobulk data. This is a question I have had, and I was pleased to see it addressed in this paper.
- * A new voom function, voomByGroup, is provided. This is a useful addition to the voom functions already available in limma. There was already a similar function for microarrays, voomaByGroup, but not for count data. This function is currently available from a GitHub repository written for this paper. I presume the intention is to add it to the limma package, and that it will be useful for both bulk and single-cell data. The function also provides a diagnostic visualization to check if group-level heteroscedasticity needs to be accounted for.

Response: We thank the reviewer for their supportive feedback on our work.

Comments and questions:

Comment 1: Is there a theoretical or practical reason not to routinely apply voomWithQualityWeights per-sample rather than per-group? Per-sample weighting should automatically include any necessary per-group weighting.

Response 1: Thank you for the excellent question. The development of the *voomWithQualityWeights* (sample-specific) variance modelling method was originally motivated by our frequent observation of outlier samples in small designed bulk RNA-seq experiments, whose presence had negative consequences in a differential expression analysis that our method could help ameliorate. In contrast, *voomQWB* or the group-specific variance trend modelling

offered in *voomByGroup* were developed to deal with situations where we see relatively higher or lower variation that is group-wide rather than driven by individual outlier samples. Each method is designed to address different scenarios and has slightly different underlying assumptions about the nature of the variation present. We are planning a follow up study to investigate the best scenarios for applying *voomWithQualityWeights* in sample-specific mode in pseudo-bulk scRNA-seq analysis. The simulation settings and types of validation datasets involved in such a study would be quite different to the ones used in our current manuscript, and as such we leave this as future work.

Comment 2: Page 3 lines 29-31, sentence "For studies involving mice...", the phrasing is odd.

Response 2: We have modified this sentence to the following:

"In studies of mouse lung tissue [30] and Xenopus tail [31], we observed some minor differences in group-specific voom trends—although, with the curves sitting close together and mostly overlapping one another (Figure 1a-b)."

Comment 3: This is an incremental extension to previous work on limma, voom, and edgeR. Someone unfamiliar with these methods might find it hard going. For example, what is the quantity "Biological Coefficient of Variation"? There is a nice description of the voom method and variants in the "Methods" section of the paper that would make the results section clearer if read first. The introduction and results sections could refer confused readers to specific relevant sections in the methods.

Response 3: We thank the reviewer for their valuable comments on how to improve the clarity of manuscript for the reader. We have added a description of 'Biological Coefficient of Variation' to the beginning of the Results section (page 3) as follows:

"BCV is a measure of the biological variability in gene expression between biological replicates and is frequently used to estimate the variance of gene expression in RNA-seq data [29]. Higher common BCV values correspond to increased biological variation between samples across genes based on the assumption of the negative binomial (NB) distribution in edgeR."

along with a reference to the relevant paper that provides further detail:

[29] McCarthy *et al.* (2012) Differential expression analysis of multifactor RNA-Seq experiments with respect to biological variation, *NAR*, 40(10): 4288-4297.

We have also added a brief summary of *limma*, *voom* and *edgeR* to the Introduction, in the following sentences (page 2):

"Most of the DE analysis methods applied on pseudo-bulk data in the literature are 'gold-standard' bulk DE analysis methods, including limma-voom, limma-trend [17, 18], edgeR [19], and DESeq2 [20]. Limma was developed for microarray data, assumes the log-transformed expression values

are normally distributed and employs linear modelling and empirical Bayes shrinkage to improve the stability and power of statistical tests. For RNA-seq data, voom and limma-trend were subsequently developed based on the assumption of normality of the log-transformed counts, using different strategies to address the dependence that is observed between the mean and the variance (referred to as heteroscedasticity) in this kind of data. Voom models the relationship between the mean and variance across all observations using a fitted LOWESS trend, and calculates precision weights based on the estimated trend for use in linear modelling. On the other hand, edgeR employs empirical Bayes shrinkage and was developed assuming gene-level counts follow a negative binomial distribution.”

Are the methods appropriate to the aims of the study, are they well described, and are necessary controls included? If not, please specify what is required.

Yes. Using real and simulated data, the authors convincingly demonstrate the necessity and effectiveness of the methods they describe.

Are the conclusions adequately supported by the data shown? If not, please explain

Yes. The paper is describing a new method applied to existing datasets published elsewhere. A variety of datasets are examined.

Are sufficient details provided to allow replication and comparison with related analyses that may have been performed? If not, please specify what is required.

Yes. The methods are well described and there is source code provided in a GitHub repository.

Does the work represent a significant advance over previously published studies?

Yes. See the contributions described above.

Is the paper of broad interest to others in the field, or of outstanding interest to a broad audience of biologists?

Pseudobulk differential expression testing is an important method for scRNA-seq data, because it can correctly allow for biological variability between samples while also being extremely computationally efficient. This paper provides guidance on doing this using the limma package and the new voomByGroup function, and highlights the need to think about group heteroscedasticity. This will be of broad interest to people working with scRNA-seq data.

Reviewer #2:

In this article, two new approaches voomQWB and voomByGroup are introduced to account for group variance-heteroscedasticity in DE analysis using scRNA-seq data. The motivation is very clear: the existing methods ignoring such heteroscedastic can substantially inflate type-I error rate

in DE analyses. The proposed approaches and comparison methods are well described and are easy to follow. The proposed approaches *voomQWB* and *voomByGroup* have been shown to outperform the gold standard methods *edgeR* and *limma* in both simulation studies and real data applications in the presence of group heteroscedasticity, so they could be competitors in DE analyses using scRNA-seq data. Nevertheless, I have some concerns as stated below.

Comment 1: *voomQWB* applies the existing R function *voomWithQualityWeights* in the R package *limma*, while *voomByGroup* simply extends *voom* by allowing group-specific variances. Therefore, the novelty of the two approaches is quite limited. The major contribution of this article is its empirical findings: *voomQWB* and *voomByGroup* could outperform the gold standard methods *edgeR* and *limma* in both simulation studies and real data applications in the presence of group heteroscedasticity.

Response 1: We thank the reviewer for their supportive feedback on our work. We agree that a major contribution of our study is showing that *voomQWB* and *voomByGroup* outperforms gold standard methods in the presence of group heteroscedasticity. We disagree, however, that the novelty of the two approaches is limited. We demonstrate in our article that group heteroscedasticity is prevalent across multiple scRNA-seq datasets, meaning that our methods are broadly applicable. In addition to this, both methods are not limited to scRNA-seq data, and can also be applied directly to bulk RNA-seq datasets, although we have not demonstrated the latter in our article since it is beyond the scope of our study.

The original paper and software for *voomWithQualityWeights* (Liu *et al.* (2015), NAR, 43(15):e97) was developed for dealing with outlier samples. The block argument which we use in this work has not been previously applied to datasets where heteroscedastic groups are present rather than individual outlier samples. In other words, both the method applied and the underlying statistical issue addressed by *voomQWB* differs from the original *voomWithQualityWeights* study.

Likewise, *voomByGroup* addresses heteroscedastic group-specific mean-variance trends, whilst the *voom* method fits an averaged global trend to the data. To implement this, the *voomByGroup* code was written to accommodate individual group-specific mean-variance trends, generate group-level plots, and calculate group-specific BCV values - these features are not available in *voom*. The novelty in our study of *voomByGroup* and *voomQWB* lies in their ability to address group-specific heteroscedasticity in pseudo-bulk scRNA-seq data, which to the best of our knowledge, has not been addressed previously.

Comment 2: In the homoscedastic-group situation, *limma* is generally more powerful than *voomQWB* and *voomByGroup* with comparable type-I error rates. This is because *voomQWB* and *voomByGroup* need to estimate more variance parameters than *limma*. Therefore, *voomQWB* and *voomByGroup* are preferred only when group variances differ to some extent. This raises a question: how to choose a suitable approach in practice? For example, can the authors give a cut-off of the absolute value of the BCV difference between two groups? So that *voomQWB* and *voomByGroup* are preferred against *limma* and *edgeR* if and only if the difference exceeds the cut-off.

Response 2: We thank the reviewer for their question. In a simulation with common BCV values ranging between 0.20 and 0.27, we observe that results differ for group variance methods versus standard methods (Figure 3). After analysing multiple datasets, we found that a 0.07 difference between the lowest and highest common BCV values (i.e. 35% increase or decrease in variation) is not uncommon, and in many datasets the differences exceeded this. We chose not to offer a specific BCV cut-off in our study however, as the choice between methods depends on both the differences in the level and shape of the group-wise mean-variance trends, which may not be captured by differences in common BCV values between groups alone (i.e. groups can have similar BCV values but fairly different trends). For example, in the mean-variance plot of Figure 1d, the NSIP group has a common BCV value of 0.446 and the Control group has a common BCV value of 0.488 (i.e 9% difference) but their mean-variance trend curves are very distinct from each other. This means that precision weights assigned to the counts in each group will be quite different from each other, especially in the high and low count range where the curves are most different. For this reason, we recommend users begin by running *voomByGroup* so that they can visualise both the mean-variance trend plots and common BCV estimates together. We recommend using *voom* if the common BCV estimates and position and shape of the group-specific trends are similar between groups (e.g. **Figure 4b**); otherwise we would recommend *voomByGroup* or *voomQWB*. We explain these recommendations in the Discussion section (page 10) as follows:

*“We recommend the use of either *voomByGroup* or *voomQWB* over gold-standard methods in scRNA-seq pseudo-bulk analysis in datasets that exhibit heteroscedastic variation across different experimental groups. The *voomByGroup* software provides useful diagnostic plots that can help guide the choice of method, with code that is easy to run, taking similar inputs to the widely used and well-established *voom* approach (see Methods). Running *voomByGroup* first can allow the analyst to determine the level of heteroscedasticity in a given dataset. For example, if the mean-variance trends per group are mostly overlapping each other, then group variance methods are likely to offer very similar results to current gold-standard methods (Figure 4b-c). In this case, method choice will not affect the results much, and one may prefer to choose a method that is simpler, based on fewer assumptions, such as *voom*. If *voomByGroup* mean-variance plots show distinct trends in one or more groups, then the variance for that group can be more closely modelled using *voomByGroup* or *voomQWB* (e.g. “ST40 3” in Figure 1b, “HC” in Figure 1c, and “NSIP” and “cHP” in Figure 1d). In such cases, methods that explicitly model group-specific variability are highly recommended over standard methods that do not. Moreover, the common BCV values that are automatically generated and displayed on these plots provide summary information about differences in mean dispersion for different groups calculated across all genes.”*

Comment 3: The simulation studies only cover low vs low and high vs high comparisons. What about low vs high comparison?

Response 3: We thank the reviewer for their question. The results for low versus high were displayed in **Figure 3b** in the original submission (repeated below - refer to the green points in

this plot), but were not described in detail in the results text. We have added a new paragraph to the Results (Page 7) to address this as follows:

“In the Low vs High comparison, there were no significant benefits in applying the group variance methods even though these methods provide more accurate variance estimates than the standard methods. When using voom, the overall variance trend that is applied to all samples and groups would under-estimate the variability of high variance groups, and over-estimate this for low variance groups. When pairwise comparisons are made between High and Low groups, the over- and under-estimated values balance each other out, giving results that are similar to the more precise estimates from voomByGroup and voomQWB.”

Figure 3b

Comment 4: Only a small group size of 3 was considered in the simulation studies. It would be of interest to consider larger sample sizes to cover more real-world situations.

Response 4: We thank the reviewer for this suggestion. We examined scRNA-seq datasets with multiple conditions and samples and noticed that few have group sizes greater than 3, which may be due to the high cost of running multi-sample scRNA-seq experiments. For this reason, we believe that having 3 replicates is reasonable for our simulations. Larger sample sizes may be more common in bulk RNA-seq datasets, so if the focus were on this technology, it would be reasonable to consider including simulations with $n > 3$. While we have not made changes to our manuscript, we did conduct extra simulations using $n = 6$ samples per group to further explore its impact on the results (see additional figure below, which is similar to **Figure 3b** except with $n = 6$ samples per group, with common BCV values of 0.20, 0.22, 0.26, and 0.28 per group). The results from this simulation with increased sample size are broadly similar to those observed in **Figure 3b**.

Comment 5: The numerical studies did not include the existing DE analysis methods specifically tailored for scRNA-seq data, such as MAST, BPSC, and DEsingle. Why?

Response 5: We thank the reviewer for their suggestion to include additional DE methods tailored to scRNA-seq data. The reason why we did not look into such methods is that recent benchmarking studies, using bulk DE methods on pseudobulk samples show consistently better performance than the tailored scRNA-seq methods in multi-sample DE analysis. We mention this in the Introduction (pages 1-2) as follows:

“Another analysis strategy performs DE analysis on pseudo-bulk samples that are created by cell aggregation [12]. This strategy was pointed out to perform better than single-cell methods that treat each cell as an independent replicate in the analysis in two independent studies [13, 14]. Through the use of an aggregation approach, dependencies between cells from the same sample are avoided [15] so that the intrinsic variability of biological replicates is well-estimated leading to fewer false discoveries compared to methods that fail to account for this [14]. Although a generalized linear mixed model with a random effect to take care of zeros and correlation structure within a sample provides slightly more power compared to pseudo-bulk aggregation methods [13, 16], it brings a much heavier computational burden [13, 14].”

We also mention this in the Discussion (page 11) as follows:

“In our article, we focus on DE analysis of scRNA-seq pseudo-bulk data because recent benchmark studies have shown that it gives better results relative to analysing scRNA-seq data in its original non-aggregated form [13, 14, 37]. However, it is worth noting that by aggregating single-cell data to obtain pseudo-bulk samples, the variance between cells of the same sample is masked. Thus it may be useful to check cell-level gene expression and its variability, especially for any genes that are detected as significant. To account for this, Zimmerman et al. modelled the correlation structure between cells using a generalized mixed model where individuals were assigned as a random effect [16]. A similar approach was taken in Crowell et al. [13]. In a similar way, linear mixed modelling may also be accessible by using the voomQWB method together with the duplicateCorrelation function in the limma package.”

Comment 6: A couple of typos: In Supplementary Figure S9(a) and the corresponding caption, VQWB and voomQW should read voomQWB.

Response: Thank you for pointing out these typos, which we have corrected in the revised version of the manuscript.

Second round of review

Reviewer 1

My initial review recommended accepting the paper with minor revisions. The authors have addressed my comments in their reply and made appropriate changes to the text, improving its readability for people not closely familiar with the various methods discussed. The authors have also addressed the comments of the other reviewer. I recommend that this revised version be accepted for publication.

Reviewer 2

I am glad that some of my comments have been addressed, but my concern about the method novelty is still there. The approach `voomWithQualityWeights` developed for handling sample outliers can be directly used to handle group heterogeneity with a block design as the input of `voomWithQualityWeights`. Also, `voomByGroup` simply estimates group-specific variances with `voom`. Therefore, it might not be a good idea to assert that the two approaches `voomByGroup` and `voomWithQualityWeights` are "new".